# Effect of Leaf Grade on Taste and Aroma of Shaken Hunan Black Tea

**DOI:** 10.3390/foods13010042

**Published:** 2023-12-21

**Authors:** Kuofei Wang, Yangbo Xiao, Nianci Xie, Hao Xu, Saijun Li, Changwei Liu, Jianan Huang, Shuguang Zhang, Zhonghua Liu, Xia Yin

**Affiliations:** 1Key Laboratory of Tea Science of Ministry of Education, Hunan Agricultural University, Changsha 410128, China; wangkuofei2023@163.com (K.W.);; 2National Research Center of Engineering and Technology for Utilization of Botanical Functional Ingredients, Changsha 410128, China; 3Department of Tea Quality Chemistry and Nutrition Health, Tea Research Institute, Hunan Academy of Agricultural Sciences, Hunan Tea Plant and Tea Processing Observation Station of Ministry of Agriculture, Changsha 410125, China; 4Co-Innovation Center of Education Ministry for Utilization of Botanical Functional Ingredients, Changsha 410128, China; 5Key Laboratory for Evaluation and Utilization of Gene Resources of Horticultural Crops, Ministry of Agriculture and Rural Affairs of China, Hunan Agricultural University, Changsha 410128, China

**Keywords:** shaken Hunan black tea, leaves grade, taste, aroma, HS-SPME/GC-MS

## Abstract

Shaken Hunan black tea is an innovative Hunan black tea processed by adding shaking to the traditional Hunan black tea. The quality of shaken black tea is influenced by leaf grades of different maturity. In this study, the taste and aroma quality of shaken Hunan black tea processed with different grades were analyzed by sensory evaluation (SP, HPLC, and HS-SPME/GC-MS). The results showed that shaken Hunan black tea processed with one bud and two leaves has the best quality, which has a sweet, mellow, and slightly floral taste, as well as a floral, honey, and sweet aroma. Moreover, caffeine and EGCG were identified as the most important bitter and astringent substances in shaken Hunan black. Combined with the analysis of GC-MS and OAV analysis, geraniol, jasmone, β-myrcene, citral, and trans-β-ocimene might be the most important components that affect the sweet aroma, while methyl jasmonate, indole, and nerolidol were the key components that affect the floral aroma of shaken Hunan black tea. This study lays a foundation for this study of the taste and aroma characteristics of shaken Hunan black tea and guides enterprises to improve shaken black tea processing technology.

## 1. Introduction

Black tea has become one of the most popular beverages in the world because of its unique taste and aroma [1]. Hunan black tea, a representative Chinese black tea, is renowned for its distinctive nectar aroma and sweet taste [2,3,4]. The formation of these qualities in Hunan black tea is influenced by the process and raw materials. In recent years, innovative technology has been introduced to improve the quality of Hunan black tea, specifically through the incorporation of the shaking process during withering, which is the first and crucial step [5]. Shaking involves continuously squeezing, rubbing, and colliding the fresh leaf blades, leading to noticeable modifications in leaf morphology, color, aroma, and biochemical composition [6]. These alterations ultimately contribute to the enhancement of taste and aroma [7,8,9].

The quality of Hunan black tea is influenced by the leaf grade, which determines the raw material quality. Maturity affects the quality of black tea by regulating the metabolites mainly through the types and proportions of components contained in fresh leaves [10]. Studies have shown that the leaf grades had a significant impact on the quality of ancient black tea [11]. Furthermore, research by Yin found notable variations in the quality of standard samples of Hunan black tea with different leaf grades [12]. It is worth noting that the shaking process requires tea leaves to have a certain maturity. Only tea leaves that have reached a certain level of maturity can undergo optimal metabolic processes during shaking, resulting in the production of volatile compounds that enhance the aroma of the tea leaves [13]. Therefore, the grade of fresh leaves plays a crucial role in determining the quality of Hunan black tea produced through shaking. However, there is a lack of research on the impact of different fresh leaf grades on the quality of shaken black tea and the corresponding patterns of change.

This study utilized sensory evaluation, spectrophotometry (SP), high-performance liquid chromatography (HPLC), and headspace solid-phase microextraction/gas chromatography-mass spectrometry (HS-SPME/GC-MS) techniques to investigate the impact of varying leaf grades on the quality of shaken Hunan black tea. We analyzed the main taste substances and volatile compounds of shaken Hunan black tea made from different grades of fresh leaves. The objective was to determine the optimal maturity levels and further examine the patterns of change and differential components in shaken Hunan black tea at different grade levels. The findings aim to provide theoretical support for production and practical applications.

## 2. Materials and Methods

### 2.1. Tea Sample Preparation

The fresh leaves of *Camellia sinensis* (L.) O. Kuntze. Zhuyeqi were sourced from the tea research base of the Hunan Academy of Agricultural Sciences, located at 28.47 N, 113.35 E, on 15 April 2023. The raw materials for the preparation of black tea with shaking included one bud and one leaf (First grade), one bud and two leaves (Second grade), one bud and three leaves (Third grade), and one bud and four leaves (Fourth grade) tea leaves. A combination of traditional techniques (withering, rolling, fermentation, and drying) and shaking processes were employed in the preparation of black tea. The withering process was conducted at a temperature range of 24 to 26 °C and a humidity range of 65% to 75% for 16 h. The shaking parameters involved subjecting the tea leaves to a shaking machine operating at a speed of 20 r/min for 5 min when the leaves were at 15% water loss. The kneading process consisted of a sequence of activities, namely, 10 min of no-pressure kneading, followed by 30 min of light pressure kneading, then 10 min of heavy-pressure kneading, and finally another 10 min of no-pressure kneading, all performed in a kneading machine. The fermentation parameters consisted of maintaining a temperature range of 28~30 °C and a humidity level of 90~95% for 3.5 h. The drying process involved subjecting the material to a temperature of 120 °C for 8 min using a dryer, followed by spreading and cooling for 1 h. Subsequently, the material was dried at 80 °C for 1 h using an aroma extraction machine (Figure 1).

### 2.2. Chemicals

*N,N*-dimethylformamide, acetonitrile, acetic acid, and methanol (chromatographic grade, Shanghai National Pharmaceutical Group Reagent Co, Shanghai, China). Methanol, sodium chloride, sodium carbonate, folin-phenol, disodium hydrogen phosphate, potassium dihydrogen phosphate, ninhydrin, stannous chloride, aluminum trichloride, anthrone, anhydrous dextrose, concentrated sulfuric acid (analytical grade), Shanghai National Pharmaceutical Group Reagent Co, Shanghai China. Epicatechin (EC), epigallocatechin (EGC), catechin (D, L-C), epigallocatechin gallate (EGCG), epigallocatechin gallate (ECG), gallocatechin gallate (GCG), aspartic acid, serine, glutamic acid, glycine, histidine, arginine, threonine, alanine, proline, theanine, cysteine, tyrosine, valine, methionine, lysine, isoleucine, leucine, phenylalanine standard, n-alkane mixed standard (C7~C30), Sigma-Aldrich Corp., St. Louis, MO, USA. Ethyl decanoate (99%), Shanghai Aladdin Biochemical Technology Co, Shanghai, China.

### 2.3. Sensory Evaluation

According to the prescribed black tea evaluation procedure outlined in the “Tea Sensory Evaluation Methods” (GB/T 23776-2018) and “Terms of tea sensory tests” (GB/T 14487-2017), a quantity of 3.0 g of black tea samples was meticulously measured and placed into the designated evaluation cup. Subsequently, 150 mL of boiling water was added to the cup, and after a steeping period of 5 min, the infusion was meticulously filtered and transferred into the designated white ceramic bowl. To ensure a comprehensive assessment, a panel consisting of seven tea reviewers, ranging in age from 25 to 57 years old and possessing national professional qualification certificates, was assembled. This panel was tasked with providing detailed descriptions and assigning scores to both the aroma and taste attributes of the black tea, employing the "Terminology of Tea Evaluation” as their reference framework.

### 2.4. Analysis of Tea Polyphenol, Catechin, Gallic Acid, Alkaloid, and Free Amino Acid Content

#### 2.4.1. Analysis of Tea Polyphenols by Spectrophotometry (SP)

The tea polyphenols were detected by the spectrophotometric method. Weigh 0.200 g of tea powder in a 10 mL centrifuge tube, add 5 mL of 70% methanol, water bath at 70 °C for 10 min in a water bath, centrifuge at 3500 r/min for 10 min, take the supernatant, repeat twice, and combine the supernatant, and then fix it to 10 mL with 70% methanol to obtain the crude extract of tea polyphenols. The concentration of 10, 20, 30, 40, and 50 μg/mL gallic acid standard solution was configured. Each of 1 mL of distilled water, gallic acid standard solution, and 100-fold diluted tea polyphenol crude extract was pipetted into different stoppered test tubes. Subsequently, 5 mL of 10% folin-phenol was introduced, followed by the addition of 4 mL of 7.5% Na_2_CO_3_ solution between the time frames of 3 and 8 min after the initiation of the reaction. The reaction was then allowed to proceed at room temperature for 60 min, after which the absorbance value was measured at a wavelength of 765 nm. The construction of the standard curve involved plotting the gallic acid concentration as the independent variable (*X*) against the absorbance value as the dependent variable (*Y*). The resulting standard curve for gallic acid exhibited a linear relationship described by the equation *Y* = 86.608*X* + 0.4657, with a coefficient of determination (R^2^) equal to 0.9988.

#### 2.4.2. Analysis of Catechin, Gallic Acid, and Alkaloid by HPLC

The aqueous extract method involves weighing 3 g (with an accuracy of 0.001 g) of finely ground samples in a 500 mL conical flask. 450 mL of boiling distilled water is then added to the flask. The mixture is promptly transferred to a boiling water bath and left to immerse for 45 min, with intermittent shaking every 10 min. After completion, the mixture is immediately subjected to decompression filtration while still hot. The residue is washed 2–3 times with a small amount of hot distilled water. The resulting filtrate is transferred to a 500 mL volumetric flask, allowed to cool, and then adjusted to the scale with water. The flask is thoroughly shaken to ensure proper mixing.

The six catechin fractions (EC, EGC, D, L-C, EGCG, ECG, and GCG), three alkaloid fractions (caffeine, theobromine, and theophylline), and gallic acid were concurrently quantified using HPLC. The chromatographic parameters employed were as follows: a C18 column (4.6 mm × 150 mm, 5 μm), a detection wavelength of 278 nm, an injection volume of 10 μL, a column temperature of 30 °C, and a flow rate of 1.0 mL/min. The mobile phase A consisted of ultrapure water, while the mobile phase B was composed of a mixture of N, N-dimethylformamide, methanol, and glacial acetic acid (*N*,*N*-dimethylformamide: methanol: glacial acetic acid = 39.5:2:1.5). The separation process involved gradient elution, with mobile phase B initially at 9%, increasing to 14% after 10 min, further increasing to 36% after 27 min, maintaining this concentration for 4 min, decreasing to 9% at 32 min, and concluding at 37 min.

#### 2.4.3. Analysis of Amino Acids by HPLC

A total of 18 amino acid components were concurrently quantified using HPLC. The chromatographic conditions employed in this study included the use of a Waters ACCQ-TagTM column (3.9 mm × 150 mm, 5 μm) for separation. The detection wavelength was set at 248 nm, and an injection volume of 10 μL was used. The AccQ•Tag method is a pre-column derivatization technique for amino acid analysis of hydrolyzed peptides and proteins. The reconstituted 10 μL samples were derivatized with the AccQ-Fluor reagent kit (WAT0052881, Waters Corp., Milford, MA, USA). AccQ-Fluor borate buffer (70 μL) was added to the sample tube with a micropipette and vortexed. Thereafter, 20 μL of AccQ-Fluor reagent was added and immediately vortexed for 30 s, and the contents were transferred to maximum recovery vials. The vials were heated for 10 min in a water bath at 55 °C before the separation of amino acids using HPLC. The column temperature was maintained at 37 °C, and a flow rate of 1 mL/min was applied. The mobile phase consisted of 10% ACCQ liquid for phase A and 60% acetonitrile for phase B. A gradient elution method was employed, with the initial concentration of phase B set at 2% and gradually increased to 7% at 15 min, 10% at 19 min, 33% at 32 min, and finally reaching 100% at 34 min. This concentration was maintained for 3 min before decreasing to 0% at 39 min.

### 2.5. Analysis of Hunan Black Tea Aroma-Active Compounds

#### 2.5.1. Hunan Black Tea Volatile Compounds Extraction by Headspace Solid-Phase Microextraction (HS-SPME)

The 50/30 μm DVB/CAR/PDMS extraction head was positioned within the gas chromatograph’s inlet port and subjected to an aging process at a temperature of 270 °C for 30 min. Each tea sample, weighing 3.00 g, was carefully measured and deposited into a 250 mL beaker equipped with a magnetic rotor. Subsequently, 10 g of NaCl was added to the beaker, which was then filled with 150 mL of boiled water. A 20 μL solution of ethyl caprate at a concentration of 8.68 μg/mL, dissolved in a solvent consisting of 10% ethanol, was injected into the beaker. The extraction head was expeditiously enveloped with a sealing film and positioned on a magnetic stirring heating stage, where it was subjected to heating and stirring at a temperature of 80 °C and a stirring rate of 600 r-min-1 for 10 min. Subsequently, the aged extraction head was introduced into the headspace region of the beaker. Following 50 min of adsorption of the extractor head in the sample bottle at a temperature of 80 °C and a stirring rate of 600 r/min, the extractor head was inserted into the gas chromatograph’s inlet and thermally resolved at a temperature of 250 °C for 5 min [2,3,4].

#### 2.5.2. Qualitative and Quantitative Analysis of the Volatiles by Gas Chromatography-Mass Spectrometry (GC-MS)

The analysis was conducted using an HP-5MS capillary column with a carrier gas consisting of 99.999% helium at a flow rate of 1.7 mL/min. The sample was injected without a shunt, and the inlet temperature was set at 250 °C. The heating program consisted of initially maintaining a temperature of 50 °C for 2 min, followed by a gradual increase to 60 °C at a rate of 1 °C/min and another 2 min hold. Subsequently, the temperature was further increased to 104 °C at a rate of 2 °C/min. The overall heating procedure involved the following steps: starting at 50 °C with a 2 min hold, increasing to 60 °C at a rate of 1 °C/min with another 2 min hold, further increasing to 104 °C at a rate of 2 °C/min with a 1 min hold, increasing to 150 °C at a rate of 5 °C/min with a 1 min hold, and finally increasing to 220 °C at a rate of 15 °C/min with a 5-min hold. The quadrupole temperature was maintained at 150 °C, while the ion source temperature was set to 230 °C. The ion source utilized in this study was an EI source, characterized by an electronic energy of 70 eV and an ionization voltage of 1540 mV. The scanning range for mass-to-charge ratio (m/z) was set between 33 and 400. The NIST 2017 spectral library was employed for comparison, resulting in a matching error of 20. Additionally, the retention index was verified.

For quantitative analysis, a semi-quantitative approach was adopted using the internal standard method. The internal standard employed was 20 μL of decanoic acid ethyl ester with a concentration of 8.68 μg/mL.

### 2.6. Statistical Analysis

All measurements were performed in triplicate. The GC-MS results were subjected to data analysis using the MSD Chemstation (Agilent Technologies Inc., Palo Alto, CA, USA). The calibration curves for odorants and associated data processing were conducted utilizing Excel 2019 (Microsoft Corp., Redmond, WA, USA). Bar graphs and radar maps were generated using Origin 2023 (Originlab Corp., Northampton, MA, USA). Multivariate statistical analyses were carried out employing SPSS Statistics 20.0 (IBM Corp., Armonk, NY, USA). Principal component analysis (PCA) was conducted using Simca 14.1 (Umetrics Corp., Umea, Sweden). The heatmap was generated using Prism 9 (Microsoft Corp., Redmond, WA, USA).

## 3. Results

### 3.1. Sensory Evaluation

The results of the sensory evaluation (Table 1) showed that there were differences in the types and scores for taste and aroma of shaken Hunan black tea prepared with different grades. As maturity increased, the taste transformed from slightly astringent to sweet and mellow. The highest taste score (92.5) was obtained by the group consisting of one bud and three leaves, indicating a sweet taste. The group with one bud and two leaves followed closely. In terms of aroma, the fragrance of shaken Hunan black tea changed from tender sweet to nectar and eventually pure sweet as the maturity of fresh leaves increased. The highest aroma score (93.5) was achieved by the group with one bud and two leaves, exhibiting a nectar fragrance. The group with one bud and three leaves showed a sweet fragrance with floral notes. The groups with one bud, one bud, and four leaves had lower scores, indicating a tender and sweet fragrance. The comprehensive analysis revealed that shaken Hunan black tea with one bud and two leaves performed the best.

### 3.2. Analysis of Main Taste Compounds

Tea polyphenols, catechins, gallic acid, alkaloids, and amino acids are the primary taste components found in tea. In this study, PCA analysis was performed on Hunan black tea samples of various grades, focusing on the content of these taste components. The R^2^X value of 75.3% suggests that the model has strong predictive ability (Figure 2). The PCA diagram revealed distinct clustering and separation patterns among the different samples, indicating variations in the main taste substances of Hunan black tea derived from fresh leaves of varying grades.

The objective of this study was to examine the variations in taste substances among different grades of shaken Hunan black tea. The results (Table 2) showed a gradual decrease in the levels of tea polyphenols, catechins, gallic acid, and alkaloids as the grade of fresh leaves decreased. In contrast, there was an increase in amino acid content as the fresh leaf grade decreased, which contradicted the trend observed in conventional leaves [12]. One possible explanation for this phenomenon could be attributed to two factors. Firstly, the selection of raw materials played a role. In this study, raw materials of different grades were obtained from the same fluffy tops with the same growth potential. As a result, the lower-grade samples had a higher stem content, which is known to be rich in free amino acids [14]. Secondly, the shaking process used in this study may have also contributed to this outcome. Previous research has shown that shaking effectively reduces the levels of polyphenolics, leading to a decrease in the bitterness and astringency of black tea [8].

Additional heat map analysis was conducted on different taste components, including theobromine and caffeine, various catechins, phenylalanine, leucine, and a few other amino acids. The results indicated a decrease in the content of these compounds as the fresh leaf grade decreased. This can be attributed to the fact that alkaloids such as caffeine and theobromine are synthesized in the stem tips of tea plants, and their content gradually diminishes as the leaves and stems mature [15]. This result ties well with previous studies; secondary metabolites tend to accumulate in higher quantities in the stems or leaves of lower grades [16].

The contribution of major taste substances to the taste of tea is influenced not only by their content but also by their taste active value (TAV) (Table 3) [17]. Caffeine, ranging from 30.77 to 46.74, is a significant source of bitterness in shaken Hunan black tea (Table 3). EGCG, with a TAV ranging from 1.09 to 1.82, is an important astringent substance in shaken Hunan black tea. Gallic acid, glutamic acid, and aspartic acid, with TAVs greater than 1, have a notable impact on acidity. Catechin, gallic acid, and alkaloids contribute to the bitter and astringent taste attributes, which play a crucial role in determining the intensity of bitter and astringent tastes in tea. These attributes decrease in magnitude as the grade of the fresh leaf decreases [18]. Additionally, phenylalanine and leucine, classified as bitter amino acids, significantly contribute to tea bitterness. Conversely, theanine, glutamic acid, and threonine, categorized as sweet amino acids, act as inhibitors of the bitter and astringent taste in tea [19]. The present study observed a decrease in the bitter, astringent, and acidic tastes of shaken Hunan black tea as the fresh leaf maturity increased, which aligns with the findings of the sensory evaluation.

### 3.3. Analysis of Aroma Compounds

#### 3.3.1. Analysis of Aroma Compounds Identified by GC-MS

To gain insight into the volatile components present in shaken Hunan black tea, the volatile components were analyzed using HS-SPME/GC-MS. Through the comparison with the NIST 2017 spectral library and retention index validation, a total of 144 volatile components were identified in shaken Hunan black tea at different grade levels (refer to Table 4). Based on the metabolic pathway classification [20], the 144 volatile components can be classified into five distinct groups: fatty acid degradation volatiles (FADVs), volatile terpenoids (VTs), amino acid degradation volatiles (AADVs), carotenoids degradation volatiles (CDVs), and other classes of volatiles. The different categories of volatiles are as follows: FADVs include 61 volatiles, VTs include 34 volatiles, AADVs include 18 volatiles, CDVs include 16 volatiles, and there are 15 components that fall under the category of ‘Others’. The mean value of VTs was found to be 103.46 mg/kg, followed by FADVs at 56.93 mg/kg, and AADVs at 33.55 mg/kg. Consequently, VTs, FADVs, and AADVs emerged as the primary volatile components in shaken Hunan black tea.

The analysis of the total volatile components and the contents of each type of volatile component of different grades of shaken Hunan black tea were analyzed, and the results are depicted in Figure 3. The concentration of total volatile components exhibited a gradual decrease as the fresh leaf grade declined. Conversely, the concentrations of FADVs and CDVs initially increased and subsequently decreased, while the concentrations of AADVs and VTs displayed a gradual decrease. Notably, no significant difference was observed in the concentration of Others. Among the various grades of shaken Hunan black tea, the first grade exhibited the highest concentrations of total volatile components, AADVs, and VTs. Conversely, the third grade demonstrated the highest concentrations of ADVs and CDVs.

#### 3.3.2. Analysis of Differential Aroma Compounds

In order to examine the variations in volatile components among various grades of Hunan black tea, a Principal Component Analysis (PCA) was conducted on a total of 144 volatile components (Table 4). The obtained R^2^X value of 68.2% suggests that the model exhibits a satisfactory level of predictability. Notably, each sample of Hunan black tea, when subjected to shaking, displayed distinct separation and clustering patterns in the PCA plots, indicating the presence of disparities in the volatile components across different grades of Hunan black tea. Subsequently, the 144 volatile components were assessed for differential composition using criteria such as a fold change value ≤0.5 or ≥2 and a *p* value < 0.05. A total of 63 differential volatile components were screened, including 29 FADV components, 24 VT components, 3 CDV components, 2 AADV components, and 2 Others components.

The concentrations of alcohols, aldehydes, ketones, and CDVs in FADVs exhibited a gradual increase as the fresh leaf grade decreased, while the concentration of esters in FAVDs initially increased and then decreased. Previous research has demonstrated that the levels of fatty acids and carotenoids increase progressively with the maturity of fresh leaves. Consequently, lower grade fresh leaves contain higher amounts of precursors for FADVs and CDVs, such as linoleic acid, linolenic acid, β-carotene, and zeaxanthin, leading to an enhanced production of FADVs and CDVs during processing [21,22,23]. Notably, the ester content of FADVs in the Fourth grade exhibited a lower level compared to that of third grade and second grade Hunan black tea, which could potentially be attributed to the diminished activity of acyltransferase in the fresh leaves of Fourth grade tea [24].

Except for trans-4,8-dimethyl-1,3,7-nonatriene, trans-β-farnesene, and α-farnesene, the content of VTs decreased with decreasing fresh leaf grade, which is consistent with the results of Yin [2]. The synthesis of volatile terpenoids, specifically monoterpenes and sesquiterpenes, primarily occurs through the mevalonate metabolic pathway and the 2-methyl-D-erythritol-4-phosphate metabolic pathway. The higher levels of VTs observed in Hunan black tea with higher grades, which undergo vigorous leaf metabolism during shaking, may be attributed to this metabolic activity [25]. The levels of trans-4,8-dimethyl-1,3,7-nonatriene, trans-β-farnesene, and α-farnesene exhibited a progressive increase as the fresh leaf grade decreased. It is plausible that these three terpene volatiles stem from the degradation of carotenoids, such as phytofluene, present in tea. This degradation process can yield α-farnesene [26].

The levels of benzyl nitrile and indole exhibited a pattern of initial increase followed by a decrease as the grade decreased, with the highest concentration observed in the third grade. Benzyl nitrile is a byproduct of phenylalanine metabolism, while Indole is produced through the catalysis of 3-glycerol phosphate by tryptophan synthase [20,27]. Further investigation is required to elucidate the underlying factors contributing to the variations in benzyl nitrile and indole contents among Hunan black tea products processed using different grades of fresh leaves, despite the observed preference for increased levels of these compounds through the shaking of fresh leaves with appropriate grade leaves in the study of oolong tea.

#### 3.3.3. OAV Analysis of Volatile Compounds

The contribution of volatile components to the aroma is influenced by both their quantity in the sample and their odor activity value (OAV). Generally, volatile components with an OAV ≥ 1 are deemed to have a significant impact on the overall aroma [28]. To investigate the variations in the aroma of Hunan black tea at different grades, a total of 63 differential volatile components were analyzed using OAV. Among these, 16 important aroma components were identified, with geraniol having the highest OAV, ranging from 69.68 to 203.29. According to odor orientation, the 16 significant aroma components were categorized into 10 floral, fruity, and sweet volatile components, as well as 6 volatile components of other aroma types (Table 5).

Geraniol, jasmone, β-myrcene, citral, and trans-β-ocimene possess sweet and fruity odor characteristics, and their concentrations in Hunan black tea decreased with decreasing grades when subjected to shaking, aligning with the findings of the sensory evaluation of the aroma (Grade 1 to Grade 4: tender-sweet aroma to sweet aroma and pure aroma) [29]. Hence, the presence of geraniol, jasmonone, β-myrcene, citral, and trans-β-ocimene significantly contributes to the variations in the sweet aroma of different grades of Hunan black tea when subjected to shaking. Furthermore, the floral odor attributes of methyl jasmonate, indole, and nerolidol are noteworthy, as their concentrations in Hunan black tea exhibit a pattern of initially increasing and subsequently decreasing with decreasing grades, aligning with the outcomes of the sensory evaluation of aroma. Consequently, methyl jasmonate, indole, and nerolidol emerge as pivotal aroma active components that account for the distinctions observed among the various grades of shaken Hunan black tea [30,31].

The fragrance of geraniol exhibited a refined and pleasant rosy and sweet scent, and it demonstrated the highest OAV in first-grade black tea with agitation, suggesting its significance as a crucial aroma-active compound. Furthermore, geraniol also serves as a significant aroma component in various other tea varieties, including green tea, white tea, and oolong tea [32,33,34]. Methyl jasmonate, primarily derived from the metabolic pathway of linolenic acid, plays a pivotal role as a floral component in shaken Hunan black tea with agitation and serves as an endogenous signaling molecule in the plant’s defense response [35]. Indole, a compound formed by the fusion of pyrrole and benzene, emits a potent fecal scent at elevated concentrations; however, when significantly diluted, it emanates a floral fragrance. Moreover, indole serves as a significant floral constituent in black and oolong tea [27,36,37].

## 4. Conclusions

This study aimed to examine the impact of different grades of fresh leaves on the taste and aroma of shaken Hunan black tea. The researchers used sensory evaluation along with SP, HPLC, and HS-SPME/GC-MS techniques. The findings revealed that ‘Zhuyeqi’ leaves with one bud and two leaves were the optimal grade for processing shaken Hunan black tea. The sensory evaluation showed that the taste of shaken Hunan black tea transformed from bitter and astringent to mellow, while the aroma changed from delicate and sweet to floral and ultimately pure as the grade of fresh leaves decreased. The analysis of SP and HPLC indicated a decrease in the content of polyphenols, catechins, alkaloids, and gallic acid in shaken Hunan black tea, along with an increase in amino acids as the fresh leaf grade decreased. Caffeine and EGCG were identified as the primary contributors to its bitter and astringent taste. Through GC-MS analysis, the researchers identified 63 differential volatile components in different grades of shaken Hunan black tea. Some of the key components influencing the sweet aroma were geraniol, jasmone, β-myrcene, citral, and trans-β-ocimene, while methyl jasmonate, indole, and nerolidol were identified as key constituents contributing to the floral aroma. These findings have theoretical significance for the production and processing of shaken black tea. Nevertheless, the optimum technology and the chemical changes during processing for shaken Hunan black tea with one bud and two leaves require further research.

## Figures and Tables

**Figure 1 foods-13-00042-f001:**
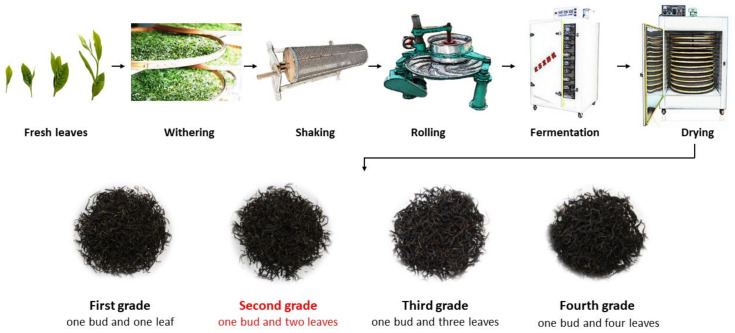
The Hunan black tea sample preparation process.

**Figure 2 foods-13-00042-f002:**
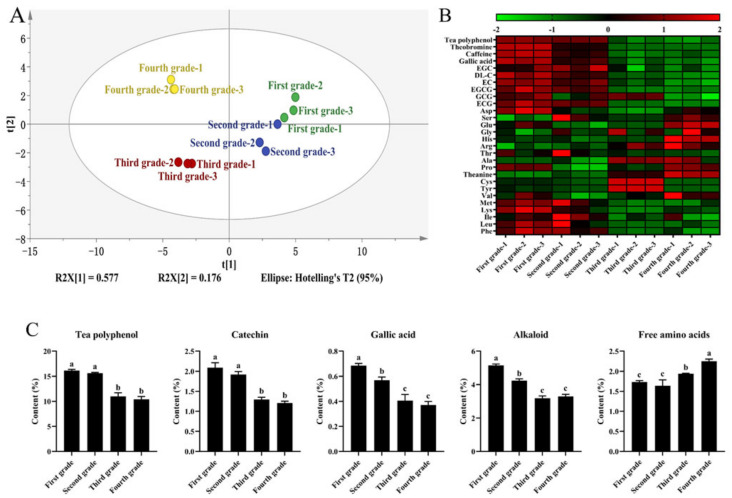
Analysis of the main taste components of shaken Hunan black tea. (**A**): PCA analysis; (**B**): thermogram analysis; (**C**): analysis of main taste components in different categories. Different low case letters above columns indicate statistical differences at *p* < 0.05.

**Figure 3 foods-13-00042-f003:**
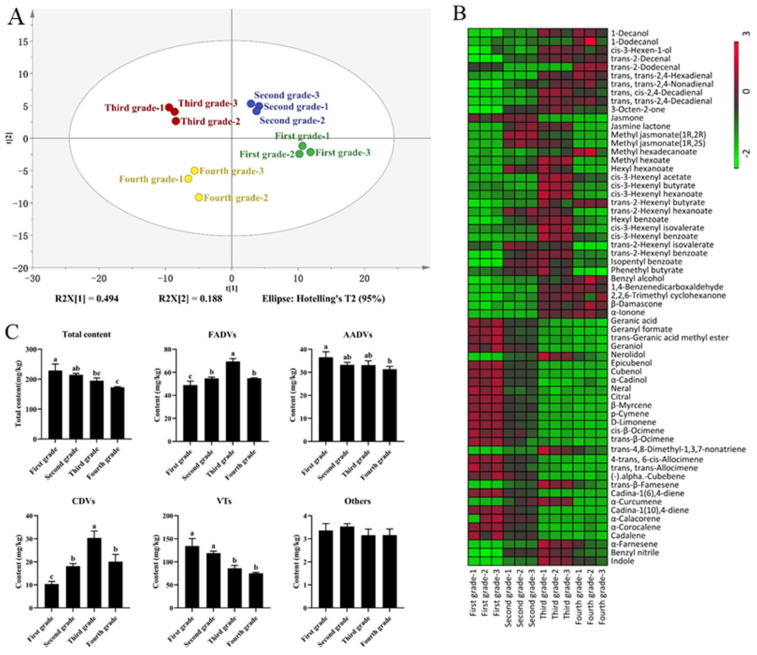
Analysis of volatile components of Hunan black tea. (**A**): PCA analysis; (**B**): heat map analysis of differential volatile components; (**C**): content of total volatile components and content of various volatile components. Different low case letters above columns indicate statistical differences at *p* < 0.05.

**Table 1 foods-13-00042-t001:** Sensory evaluation results of Hunan black tea with different leaf grades.

Grades	Taste Description	Score	Aroma Descriptions	Score
First grade	Slightly bitter	90.5	Tender sweet	90.5
Second grade	Sweet, mellow, slightly floral	92	Floral, honey, sweet	93.5
Third grade	Sweet, mellow, floral	92.5	Sweet with floral	93
Fourth grade	Mellow	90.5	Pure sweet	91

First grade: one bud and one leaf; Second grade: one bud and two leaves; Third grade: one bud and three leaves; Fourth grade: one bud and four leaves.

**Table 2 foods-13-00042-t002:** Table of the content of the main taste components (%).

Components	First Grade	Second Grade	Third Grade	Fourth Grade
Polyphenol	16.15 ± 0.25	15.67 ± 0.16	11.1 ± 0.69	10.51 ± 0.55
Theobromine	0.47 ± 0.00	0.28 ± 0.00	0.11 ± 0.00	0.1 ± 0.00
Gallic acid	0.68 ± 0.01	0.56 ± 0.02	0.4 ± 0.04	0.37 ± 0.02
CAF	4.67 ± 0.07	3.96 ± 0.09	3.07 ± 0.12	3.18 ± 0.12
EGC	0.15 ± 0.00	0.16 ± 0.00	0.12 ± 0.01	0.12 ± 0.00
D, L-C	0.09 ± 0.00	0.07 ± 0.00	0.04 ± 0.00	0.03 ± 0.00
EC	0.21 ± 0.00	0.2 ± 0.01	0.15 ± 0.00	0.13 ± 0.00
EGCG	1.09 ± 0.08	0.99 ± 0.04	0.65 ± 0.02	0.65 ± 0.02
GCG	0.13 ± 0.00	0.12 ± 0.00	0.13 ± 0.00	0.1 ± 0.00
ECG	0.4 ± 0.03	0.36 ± 0.01	0.17 ± 0.00	0.15 ± 0.00
Asp	0.24 ± 0.02	0.18 ± 0.00	0.15 ± 0.00	0.19 ± 0.01
Ser	0.08 ± 0.00	0.11 ± 0.01	0.09 ± 0.00	0.11 ± 0.00
Glu	0.21 ± 0.02	0.15 ± 0.01	0.17 ± 0.00	0.26 ± 0.01
Gly	0 ± 0.00	0.01 ± 0.00	0.01 ± 0.00	0.01 ± 0.00
His	0.06 ± 0.00	0.06 ± 0.00	0.06 ± 0.00	0.1 ± 0.00
Arg	0 ± 0.00	0.01 ± 0.00	0.02 ± 0.00	0.02 ± 0.00
Thr	0.03 ± 0.00	0.03 ± 0.00	0.03 ± 0.00	0.03 ± 0.00
Ala	0.03 ± 0.00	0.03 ± 0.00	0.04 ± 0.00	0.04 ± 0.00
Pro	0.07 ± 0.00	0.05 ± 0.00	0.06 ± 0.00	0.07 ± 0.00
Theanine	0.63 ± 0.02	0.69 ± 0.04	0.92 ± 0.00	1.14 ± 0.04
Cys	0.01 ± 0.00	0.01 ± 0.00	0.05 ± 0.02	0.01 ± 0.00
Tyr	0.04 ± 0.00	0.03 ± 0.00	0.11 ± 0.05	0.02 ± 0.00
Val	0.02 ± 0.00	0.01 ± 0.00	0.02 ± 0.00	0.02 ± 0.00
Met	0.06 ± 0.00	0.06 ± 0.00	0.04 ± 0.00	0.04 ± 0.00
Lys	0.05 ± 0.00	0.04 ± 0.00	0.03 ± 0.00	0.04 ± 0.00
Ile	0.02 ± 0.00	0.02 ± 0.00	0.02 ± 0.00	0.02 ± 0.00
Leu	0.05 ± 0.00	0.04 ± 0.01	0.03 ± 0.00	0.03 ± 0.00
Phe	0.04 ± 0.00	0.04 ± 0.00	0.03 ± 0.00	0.02 ± 0.00

**Table 3 foods-13-00042-t003:** TAV analysis of the main taste components.

Components	OT (mg/L)	First Grade	Second Grade	Third Grade	Fourth Grade
Bitterness					
D, L-C	290	0.32	0.26	0.17	0.14
EGCG	300	3.64	3.32	2.20	2.17
GCG	180	0.73	0.68	0.73	0.60
ECG	200	2.03	1.82	0.88	0.76
Gallic acid	140	4.89	4.05	2.89	2.65
CAF	100	46.74	39.65	30.77	31.87
Theobromine	144	3.31	1.95	0.78	0.71
Astringency					
DL-C	170	0.54	0.45	0.28	0.23
EGCG	600	1.82	1.66	1.10	1.09
GCG	180	0.73	0.68	0.73	0.60
ECG	500	0.81	0.73	0.35	0.30
Theanine	1050	0.60	0.67	0.88	1.09
Sourness					
Gallic acid	187	3.66	3.04	2.17	1.98
Glu	147	1.45	1.05	1.17	1.78
Asp	143	1.72	1.27	1.06	1.37
Umaminess					
Glu	440	0.48	0.35	0.39	0.59
Theanine	4200	0.15	0.17	0.22	0.27
Asp	530	0.46	0.34	0.28	0.37

Odor threshold in water (OT) is found in the literature with a database (http://www.flavornet.org/flavornet.html, accessed on 6 October 2023).

**Table 4 foods-13-00042-t004:** Identification of 144 differential volatile compounds from shaken Hunan black tea (ug/L).

No.	Compounds	RI	CAS	First Grade	Second Grade	Third Grade	Fourth Grade
FADVs							
1	1-Penten-3-ol	4.134	616-25-1	38.27 ± 0.00	46.7 ± 0.00	54.76 ± 2.66	46.27 ± 5.53
2	1-Penten-3-one	4.244	1629-58-9	50.72 ± 3.66	64.8 ± 8.7	79.41 ± 7.3	64.67 ± 15.96
3	Pentanal	4.399	110-62-3	97.93 ± 4.5	109.54 ± 0.57	105.1 ± 13.13	71.11 ± 10.4
4	trans-2-Pentenal	5.64	1576-87-0	83.49 ± 2.02	90.19 ± 10.95	105.28 ± 19.39	86.51 ± 27.02
5	cis-2-Penten-1-ol	6.021	1576-95-0	235 ± 17.16	181.99 ± 23.86	168.64 ± 7.91	157.44 ± 20.03
6	Hexanal	6.93	66-25-1	2894.05 ± 68.05	3207.76 ± 82.48	3921.23 ± 180.04	3279.69 ± 382.49
7	2-Hexenal	8.879	505-57-7	63.98 ± 4.26	67.98 ± 4.81	101.97 ± 8.83	101.06 ± 20.93
8	trans-2-Hexenal	9.216	6728-26-3	5408.76 ± 1171.89	5758.28 ± 320.82	7064.16 ± 515.94	7035.24 ± 785.45
9	cis-3-Hexen-1-ol	9.351	928-96-1	687.89 ± 359.75	823.17 ± 124.97	1418.05 ± 40.85	1221.23 ± 237.64
10	trans-2-Hexen-1-ol	9.911	928-95-0	402.69 ± 56.58	357.5 ± 23.8	271.93 ± 13.63	340.7 ± 33.52
11	1-Hexanol	10.026	111-27-3	729.55 ± 77.23	415.83 ± 28.72	535.05 ± 18.99	589.76 ± 12.51
12	Heptanal	11.939	111-71-7	2084.18 ± 21.23	2568.03 ± 14.59	2370.97 ± 75.24	1665.24 ± 199.14
13	trans, trans-2,4-Hexadienal	12.565	142-83-6	63.91 ± 11.09	51.27 ± 0	136.24 ± 10.17	162.05 ± 11.47
14	Methyl hexoate	13.642	106-70-7	7.58 ± 1.13	11.92 ± 1.79	23.13 ± 1.97	7.12 ± 0.55
15	trans-2-Heptenal	16.037	18829-55-5	172.5 ± 6.47	277.37 ± 21.05	343.39 ± 20.68	249.35 ± 33.25
16	1-Heptanol	17.364	111-70-6	176.71 ± 8.63	205.55 ± 15.41	169.34 ± 7.78	128.03 ± 22.41
17	1-Octen-3-ol	18.194	3391-86-4	210.6 ± 4.39	239.95 ± 11.84	304.4 ± 58.53	217.31 ± 21.71
18	2,3-Octanedione	18.692	585-25-1	101.94 ± 5.33	118.75 ± 17.35	179.22 ± 29.74	137.88 ± 20.49
19	Octanal	20.299	124-13-0	1020.33 ± 99.91	1139.51 ± 136.32	1141.59 ± 125.5	1006.01 ± 266.03
20	cis-3-Hexenyl acetate	20.736	3681-71-8	0 ± 0	0 ± 0	400.73 ± 31.8	169.62 ± 52.73
21	trans,trans-2,4-Heptadienal	21.005	4313-03-5	223.98 ± 9.36	315.97 ± 50.7	728.05 ± 15.08	546.2 ± 79.35
22	3-Octen-2-one	23.6	1669-44-9	0 ± 0	63.9 ± 0.00	81.51 ± 8.29	53.81 ± 0.62
23	trans-2-Octenal	25.191	2548-87-0	1022.37 ± 87.04	1378.98 ± 69.84	1780.4 ± 43.89	1315.18 ± 168.75
24	1-Octanol	26.445	111-87-5	1106.92 ± 9.8	1331.1 ± 241.48	1364.52 ± 268.83	1169.99 ± 77.66
25	Nonanal	29.345	124-19-6	7676.34 ± 532.51	7036.87 ± 632.12	8467.13 ± 907.42	8714.89 ± 102.77
26	4-trans, 6-cis-Allocimene	31.206	7216-56-0	511.07 ± 15.36	368.52 ± 16.55	188.04 ± 13.97	144.53 ± 2.21
27	trans, trans-Allocimene	32.156	3016-19-1	296.81 ± 38.17	207.66 ± 21.39	127.92 ± 24.39	114.53 ± 1.09
28	trans-2-Nonenal	33.604	18829-56-6	969.19 ± 198.29	966.24 ± 134.15	1162.87 ± 253.65	1004.42 ± 35.31
29	1-Nonanol	34.672	143-08-8	2198.38 ± 63.1	2511.43 ± 154.16	2294.53 ± 186.04	2358.94 ± 70.44
30	cis-3-Hexenyl butyrate	35.887	16491-36-4	1536.18 ± 118.74	1198.19 ± 110.67	3278.74 ± 170.71	1415.61 ± 81.5
31	Hexyl butanoate	36.222	2639-63-6	495.64 ± 79.9	563.45 ± 0.00	726.68 ± 112.53	451.72 ± 23.07
32	trans-2-Hexenyl butyrate	36.529	53398-83-7	373.22 ± 40.96	530.01 ± 0.15	726.31 ± 143.54	798.08 ± 0.00
33	Decanal	37.208	112-31-2	4989.41 ± 1584.03	4196.26 ± 754.43	5238.1 ± 1083.9	5995.52 ± 1901.05
34	trans,trans-2,4-Nonadienal	37.718	5910-87-2	0 ± 0	136.57 ± 32.02	176.62 ± 34.32	209.83 ± 34.17
35	cis-3-Hexenyl isovalerate	39.268	35154-45-1	1998.89 ± 194.84	2257.7 ± 57.42	4563.56 ± 464.52	1519.42 ± 225.9
36	trans-2-Hexenyl isovalerate	39.717	68698-59-9	278.61 ± 121.03	557.97 ± 60.58	442.32 ± 54.56	0 ± 0
37	trans-2-Decenal	41.176	3913-81-3	0 ± 0	714.96 ± 30.03	1121.54 ± 41.13	1023.69 ± 157.47
38	1-Decanol	41.756	112-30-1	133.95 ± 3.75	205.25 ± 7.82	304.29 ± 41.05	331.96 ± 19.15
39	trans,cis-2,4-Decadienal	42.812	25152-83-4	195.49 ± 12.43	236.26 ± 19.6	391.09 ± 9.17	321.25 ± 8.61
40	Undecanal	43.494	112-44-7	500.67 ± 149.58	423.34 ± 76.1	462.87 ± 99.68	507.06 ± 58.36
41	trans,trans-2,4-Decadienal	43.884	25152-84-5	422.78 ± 35.49	649.08 ± 98.62	1294.58 ± 76.27	1309.81 ± 73.23
42	trans-2-Undecenal	45.853	53448-07-0	581.7 ± 44.29	597.19 ± 58.97	547.05 ± 13.28	573.09 ± 38.13
43	cis-3-Hexenyl hexanoate	46.556	31501-11-8	2454.61 ± 236.61	2969.02 ± 219.66	6076.67 ± 439.71	2410.64 ± 186.57
44	Jasmone	47.206	488-10-8	3985.68 ± 477.47	4287.64 ± 10.36	2353.89 ± 83.99	2171.62 ± 19.01
45	Jasmine lactone	50.152	25524-95-2	0 ± 0	210.86 ± 53.95	210.06 ± 15.94	0 ± 0
46	Hexyl hexanoate	46.735	6378-65-0	631.9 ± 87.81	1346.62 ± 161.09	1358.15 ± 213.38	593.59 ± 0.00
47	trans-2-Hexenyl hexanoate	46.85	53398-86-0	647.55 ± 97.66	1201.48 ± 133.68	1100.87 ± 52.89	578.93 ± 39.62
48	Dodecanal	47.496	112-54-9	267.74 ± 110.52	295.54 ± 131.22	296.43 ± 73.69	381.1 ± 32.94
49	Isopentyl benzoate	48.461	94-46-2	0 ± 0	58.69 ± 3.11	54.12 ± 11.23	29.62 ± 8.33
50	Phenethyl butyrate	48.635	103-52-6	238.25 ± 18.96	322.71 ± 5.51	346.53 ± 64.7	114.19 ± 4.86
51	trans-2-Dodecenal	49.409	20407-84-5	68.75 ± 0.41	38.14 ± 0	38.14 ± 1.97	92.9 ± 2.25
52	1-Dodecanol	49.628	112-53-8	78.93 ± 24.85	94.1 ± 15.28	105.37 ± 4.08	202.3 ± 59.16
53	Tridecanal	50.497	10486-19-8	205.2 ± 69.06	185.89 ± 24.47	207.26 ± 43.97	297.45 ± 33.98
54	cis-3-Hexenyl benzoate	51.663	25152-85-6	177.2 ± 25.73	518.02 ± 72.89	2071.25 ± 207.01	1071.99 ± 56.64
55	Hexyl benzoate	51.773	6789-88-4	38.81 ± 8.95	139.67 ± 23.59	200.4 ± 4.67	106.68 ± 12.8
56	trans-2-Hexenyl benzoate	51.88	76841-70-8	0 ± 0	76.57 ± 13.54	100.24 ± 2.29	42.01 ± 1.98
57	2-Phenethyl hexanoate	52.689	6290-37-5	27.86 ± 3.79	43.81 ± 4.2	43.5 ± 7.75	49.99 ± 0.00
58	Methyl jasmonate(1R,2R)	52.772	1211-29-6	0 ± 0	547.4 ± 1.77	224.72 ± 22.65	0 ± 0
59	Methyl jasmonate(1R,2S)	53.147	39924-52-2	0 ± 0	49.08 ± 5.11	38.04 ± 6.57	0 ± 0
60	Isopropyl myristate	54.693	110-27-0	19.58 ± 4.46	26.83 ± 9.43	18.9 ± 3.58	38.6 ± 19.1
61	Methyl hexadecanoate	55.932	57-10-3	54.26 ± 10.25	80.01 ± 27.98	95.62 ± 8.93	167.17 ± 13.44
AADVs							
62	2-Methyl propanal	2.983	78-84-2	65.35 ± 24.22	98.92 ± 11.96	80.84 ± 24.53	89.88 ± 22.18
63	3-Methyl butanal	3.797	590-86-3	487.74 ± 124.05	543.12 ± 148.33	448.11 ± 17.46	377 ± 26.3
64	2-Methyl butanal	3.917	96-17-3	394.29 ± 117.71	432.4 ± 118.38	311.89 ± 8.63	272.4 ± 10.78
65	Benzaldehyde	16.266	100-52-7	2051.14 ± 355.55	2810.96 ± 124.05	2346.21 ± 301.8	2590.6 ± 42.11
66	Benzyl alcohol	23.259	100-51-6	384.99 ± 87.15	571.71 ± 6.43	763 ± 62.49	1022.63 ± 187.89
67	Benzeneacetaldehyde	23.826	122-78-1	4675.41 ± 568.68	5593.84 ± 541.02	4526.29 ± 235.04	4297.01 ± 137.86
68	Benzyl nitrile	31.921	140-29-4	109.44 ± 19.03	523.78 ± 7.62	624.94 ± 55.39	398.5 ± 48.6
69	Phenylethyl Alcohol	29.866	60-12-8	2568.63 ± 168.63	2337.5 ± 165.28	1717.27 ± 27.81	1814.17 ± 227.93
70	1,4-Benzenedicarboxaldehyde	34.605	623-27-8	157.87 ± 8.8	190.44 ± 18.18	402.91 ± 21.88	438.3 ± 3.15
71	Indole	42.756	120-72-9	122.35 ± 10.48	614.79 ± 51.73	1366.17 ± 141.78	923.86 ± 6.15
72	Methyl salicylate	36.313	119-36-8	22,761.73 ± 1261.96	15,596.99 ± 783.32	12,775.35 ± 723.05	16,918.32 ± 312.18
73	cis-3-Hexenyl isovalerate	39.268	35154-45-1	1998.89 ± 194.84	2257.7 ± 57.42	4563.56 ± 464.52	1519.42 ± 225.9
74	trans-2-Hexenyl isovalerate	39.717	68698-59-9	278.61 ± 121.03	557.97 ± 60.58	442.32 ± 54.56	0 ± 0
75	Isopentyl benzoate	48.461	94-46-2	0 ± 0	58.69 ± 3.11	54.12 ± 11.23	29.62 ± 8.33
76	Phenethyl butyrate	48.635	103-52-6	238.25 ± 18.96	322.71 ± 5.51	346.53 ± 64.7	114.19 ± 4.86
77	cis-3-Hexenyl benzoate	51.663	25152-85-6	177.2 ± 25.73	518.02 ± 72.89	2071.25 ± 207.01	1071.99 ± 56.64
78	Hexyl benzoate	51.773	6789-88-4	38.81 ± 8.95	139.67 ± 23.59	200.4 ± 4.67	106.68 ± 12.8
79	trans-2-Hexenyl benzoate	51.88	76841-70-8	0 ± 0	76.57 ± 13.54	100.24 ± 2.29	42.01 ± 1.98
CDVs							
80	6-Methyl-5-hepten-2-one	18.909	110-93-0	247.3 ± 5.66	289.98 ± 9.84	483.91 ± 91.69	267.76 ± 93.37
81	2,2,6-Trimethyl cyclohexanone	22.949	2408-37-9	132.84 ± 3.47	174.37 ± 13.96	276.01 ± 13.45	232.36 ± 42.31
82	trans-4,8-Dimethyl-1,3,7-nonatriene	30.205	19945-61-0	0 ± 0	90.98 ± 0.82	312.58 ± 77.37	167.34 ± 59.33
83	Citronellal	33.13	106-23-0	1180.57 ± 80.22	910.94 ± 122.74	623.51 ± 93.68	567.57 ± 52.65
84	Safranal	36.584	116-26-7	187.5 ± 50.41	274.18 ± 32.63	293.6 ± 0.04	377.43 ± 86.38
85	β-Cyclocitral	38.123	432-25-7	1122.94 ± 90.44	1146.58 ± 92.46	2127.92 ± 136.23	1687.48 ± 291.6
86	β-Homocyclocitral	40.713	472-66-2	604.55 ± 9.35	515.11 ± 88.05	807.02 ± 14.74	849.77 ± 28.34
87	Theaspirane	43.8	36431-72-8	311.74 ± 63.36	359.28 ± 74.56	381.4 ± 11.06	427.38 ± 34.42
88	β-Damascenone	46.665	23726-93-4	309.2 ± 89.91	294.15 ± 29.91	233.86 ± 26.16	336.4 ± 74.17
89	β-Damascone	47.736	35044-68-9	49.78 ± 11.13	58.07 ± 7.88	96.11 ± 8.01	109.88 ± 18.44
90	α-Ionone	48.176	127-41-3	177.38 ± 9.38	196.85 ± 8.48	421.87 ± 6.79	474.46 ± 1.22
91	Geranylacetone	49.029	689-67-8	1499.34 ± 557.54	1351.96 ± 415.64	1240 ± 398.83	1272.35 ± 288.62
92	β-Ionone	50.003	14901-07-6	2512.01 ± 69.71	2521.05 ± 88.64	4279.36 ± 144.72	4855.04 ± 406.99
93	trans-β-Famesene	49.154	18794-84-8	112.87 ± 25.14	619.07 ± 69.77	1244.12 ± 174.83	603.3 ± 129.38
94	α-Farnesene	50.477	502-61-4	98.24 ± 24.65	676.15 ± 38.34	1911.82 ± 302.69	1168.4 ± 376.38
95	Nerolidol	51.518	7212-44-4	1822.7 ± 436.83	8667.35 ± 500.82	15,624.83 ± 2705.14	8339.53 ± 1488.67
VTs							
96	β-Myrcene	19.274	123-35-3	5601.64 ± 74.59	4182.84 ± 120.66	2394.62 ± 190.39	2064.58 ± 114.79
97	α-Terpinen	21.391	99-86-5	161.17 ± 12.75	129.04 ± 16.2	124.79 ± 2.98	90.09 ± 1.37
98	p-Cymene	22.126	99-87-6	245.34 ± 18.56	159.14 ± 10.54	68.01 ± 3.69	51.29 ± 0.7
99	D-Limonene	22.471	138-86-3	1649.17 ± 49.87	1121.64 ± 92.15	533.74 ± 43.52	468.36 ± 46.8
100	cis-β-Ocimene	23.532	3338-55-4	1276.29 ± 126.6	930.13 ± 139.46	393.41 ± 53.05	380.73 ± 8.21
101	trans-β-Ocimene	24.414	3779-61-1	2321.92 ± 72.5	1607.66 ± 332.16	1177.06 ± 130.06	894.15 ± 71.25
102	cis-Linalool furan oxide	26.409	5989-33-3	926.3 ± 340.43	1178.89 ± 192.31	966.05 ± 100.62	940.46 ± 192.71
103	trans-Linalool furan oxide	27.77	34995-77-2	3092.22 ± 618.16	3246.97 ± 257.02	2601.84 ± 261.04	2529.07 ± 226.52
104	Linalool	26.409	5989-33-3	29,210.41 ± 6742.04	23,250.02 ± 3621.37	21,810.21 ± 2844.52	24,134.53 ± 1868.92
105	Hotrienol	29.352	29957-43-5	2468.22 ± 1059.47	3114.78 ± 935.31	3656.59 ± 1761.41	1867.19 ± 293.02
106	trans-Linalool pyran oxide	34.273	39028-58-5	202.69 ± 71.83	272.12 ± 40.23	225.47 ± 28.68	245.15 ± 32.49
107	cis-Linalool pyran oxide	34.814	14049-11-7	505.68 ± 106.71	576.28 ± 22.43	270.29 ± 44.89	540.49 ± 61.2
108	α-Terpineol	35.986	98-55-5	361.01 ± 69.48	464.2 ± 42.25	237.9 ± 8.49	274.08 ± 10.37
109	Nerol	38.911	106-25-2	2829.18 ± 97.33	2262.83 ± 142.36	1636.24 ± 175.23	1532.3 ± 12.13
110	Neral	39.758	106-26-3	1488.81 ± 95.19	929.03 ± 77.49	593.78 ± 111.11	606.35 ± 20.98
111	Geraniol	41.158	106-24-1	67,087.28 ± 11,666.53	56,246.97 ± 2333.79	26,082.64 ± 1943.35	24,305.82 ± 1231.03
112	Citral	38.123	432-25-7	4275.68 ± 136.24	3021.62 ± 232.36	1790.08 ± 168.96	1576.52 ± 13.7
113	Geranyl formate	43.283	105-86-2	190.59 ± 15.96	128.68 ± 8.32	81.56 ± 4.73	61.62 ± 1.64
114	Geranic acid	44.257	1189-09-9	652.97 ± 56.88	492.08 ± 16.5	283.48 ± 51.45	230.14 ± 10.38
115	trans-Geranic acid methyl ester	44.257	1189-09-9	488.81 ± 36.87	378.32 ± 3.46	273.8 ± 9.9	246.93 ± 19.22
116	.alpha.-Cubebene	45.294	17699-14-8	200.32 ± 26.21	170.91 ± 14.77	77.71 ± 4.05	68.22 ± 5.16
117	trans-β-Famesene	49.154	18794-84-8	112.87 ± 25.14	619.07 ± 69.77	1244.12 ± 174.83	603.3 ± 129.38
118	Cadina-1(6),4-diene	49.692	16729-00-3	256.83 ± 10.7	169.35 ± 13.58	49.99 ± 6.5	45.58 ± 0.44
119	α-Curcumene	49.904	644-30-4	0 ± 0	83.8 ± 27.88	130.41 ± 13.94	0 ± 0
120	α-Farnesene	50.477	502-61-4	98.24 ± 24.65	676.15 ± 38.34	1911.82 ± 302.69	1168.4 ± 376.38
121	Nerolidol	51.518	7212-44-4	1822.7 ± 436.83	8667.35 ± 500.82	15,624.83 ± 2705.14	8339.53 ± 1488.67
122	Cadina-1(10),4-diene	50.841	483-76-1	3260.38 ± 239.17	2339.7 ± 242.27	838.14 ± 71.55	742.53 ± 18.95
123	α-Calacorene	51.148	21391-99-1	257.26 ± 172.75	259.82 ± 8.3	95.22 ± 3.1	110.2 ± 23.93
124	Cedrol	52.246	77-53-2	116.58 ± 22.45	139.03 ± 7.32	84.96 ± 4.1	133.49 ± 24.37
125	α-Corocalene	52.483	20129-39-9	72.66 ± 3.4	52.97 ± 3.49	22.66 ± 1.5	28.81 ± 6.65
126	Epicubenol	52.578	19912-67-5	1347.8 ± 85.75	790.07 ± 25.25	193 ± 11.33	218.62 ± 39.48
127	Cubenol	52.769	21284-22-0	1388.76 ± 71.41	890.95 ± 10.28	323.3 ± 14.88	254.39 ± 49.41
128	α-Cadinol	52.917	481-34-5	228.39 ± 25.89	157.53 ± 0.46	55.21 ± 8.34	99.08 ± 25.39
129	Cadalene	53.167	483-78-3	93.54 ± 19.03	72.85 ± 4.15	35.81 ± 3.79	32.46 ± 6.69
Others							
130	Dimethyl sulfide	2.882	75-18-3	728.39 ± 239.78	640.38 ± 118.1	401.42 ± 7.37	363.06 ± 96.35
131	Dimethyl disulfide	5.389	624-92-0	13.68 ± 3.02	14.08 ± 0	12.04 ± 2.14	15.55 ± 0.7
132	3-Methyl furan	3.304	930-27-8	13.18 ± 0	16.41 ± 1.02	11.33 ± 3.34	7.94 ± 3.39
133	2-Ethylfuran	4.463	3208-16-0	156.65 ± 14.54	168.18 ± 10.4	310.92 ± 6.05	280.74 ± 41.37
134	Dimethyl trisulfide	16.964	3658-80-8	23.59 ± 0	25.27 ± 0	26.96 ± 3.43	42.02 ± 5.93
135	cis-2-(2-Pentenyl) furan	20.156	70424-13-4	126.56 ± 16.27	167.23 ± 4.95	249.88 ± 18.19	236.7 ± 4.54
136	2,6-Dimethyl-5-heptenal	24.86	106-72-9	99.94 ± 7.26	128.7 ± 5.04	115.37 ± 10.8	70.41 ± 9.83
137	5-Ethyl-6-methyl-3-hepten-2-one	32.582	57283-79-1	619.18 ± 50.3	643.18 ± 7.74	555.42 ± 44.29	410.86 ± 30.6
138	Dodecane	36.767	112-40-3	658.1 ± 32.76	694.94 ± 32	656.95 ± 62.02	636.74 ± 70.71
139	Tridecane	43.139	629-50-5	76.76 ± 9.51	71.18 ± 2.07	75.94 ± 3.6	81.51 ± 0.67
140	Tetradecane	47.176	629-59-4	170.18 ± 14.54	198.93 ± 1.63	165.92 ± 53.01	228.17 ± 4.36
141	5-Methyl-2-phenyl-2-hexenal	50.083	21843-92-4	71.88 ± 6.51	62.95 ± 4.62	38.98 ± 2.97	71.38 ± 10.66
142	2,4-Di-tert-butylphenol	50.55	96-76-4	254.43 ± 40.08	215.72 ± 14.11	145.47 ± 22.86	193.87 ± 41.96
143	Kodaflex txib	52.066	6846-50-0	269.45 ± 111.11	435.35 ± 145.68	327.52 ± 124.74	301.04 ± 73.91
144	Dibutyl phthalate	56.595	84-74-2	77.51 ± 35.85	50.01 ± 3.57	54.24 ± 28.51	59.58 ± 2.15

**Table 5 foods-13-00042-t005:** OAV analysis of differential volatile compounds.

Compounds	CAS	Odor Description	OT (μg/L)	First Grade	Second Grade	Third Grade	Fourth Grade
Floral, fruity, and sweet aroma							
Geraniol	106-24-1	Rose, floral, sweet	6.6	203.29 ± 28.87	170.45 ± 5.77	79.04 ± 4.81	69.68 ± 6.02
β-Damascone	35044-68-9	Honey, grape, fruity, floral	0.09	11.06 ± 2.02	12.90 ± 1.43	21.36 ± 1.45	23.94 ± 2.46
Jasmone	488-10-8	Jasmine, floral, sweet	7	11.39 ± 1.11	12.25 ± 0.02	6.73 ± 0.20	5.96 ± 0.34
β-Myrcene	123-35-3	Citrus, fruity, herbal	16.6	6.75 ± 0.07	5.04 ± 0.12	2.89 ± 0.19	2.56 ± 0.13
α-Ionone	127-41-3	Sweet, floral, violet	3.8	<1	1.04 ± 0.04	2.22 ± 0.03	2.34 ± 0.22
Nerolidol	7212-44-4	Woody, floral	120	<1	1.44 ± 0.07	2.60 ± 0.37	1.18 ± 0.33
Indole	120-72-9	Animal, fecal, floral	11	<1	1.12 ± 0.08	2.48 ± 0.21	1.54 ± 0.19
Citral	5392-40-5	Lemon, sweet	40	2.14 ± 0.06	1.51 ± 0.09	<1	<1
Methyl jasmonate(1R,2R)	1211-29-6	Jasmine, floral, sweet	3	—	3.65 ± 0.01	1.50 ± 0.12	—
trans-β-Ocimene	3779-61-1	Citrus, herb, sweet, warm	34	1.37 ± 0.03	<1	<1	<1
Others aroma							
trans, trans-2,4-Decadienal	25152-84-5	Oily, fatty, cucumber	0.2	42.28 ± 2.90	64.91 ± 8.05	129.46 ± 6.23	123.59 ± 11.28
trans, cis-2,4-Decadienal	25152-83-4	Fatty, green, waxy	0.07	55.85 ± 2.90	67.50 ± 4.57	111.74 ± 2.14	88.37 ± 5.04
trans, trans-2,4-Nonadienal	5910-87-2	Oil, fatty, green	0.1	—	27.31 ± 5.23	35.32 ± 5.60	38.34 ± 6.47
trans-2-Decenal	3913-81-3	Fatty, tallow, dried fish	5	—	2.86 ± 0.10	4.49 ± 0.13	4.09 ± 0.36
cis-3-Hexenyl isovalerate	35154-45-1	Fresh, green apple, fruity	20	2.00 ± 0.16	2.26 ± 0.05	4.56 ± 0.38	1.47 ± 0.15
trans-2-Dodecenal	20407-84-5	Citrus, coriander, fat, green	1.4	<1	<1	<1	1.33 ± 0.02

Odor description found in the literature with the database (http://db.foodmate.net/, accessed on 6 October 2023). OT was found in the literature with a database (http://www.flavornet.org/flavornet.html, accessed on 6 October 2023)). ‘—’ Not detected.

## Data Availability

Data are contained within the article.

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
