# Peer review of "Effect of Leaf Grade on Taste and Aroma of Shaken Hunan Black Tea"

_foods, 2023, doi:10.3390/foods13010042_

Round 1
Reviewer 1 Report
Comments and Suggestions for Authors
The topic of the study was very original and it has suitable references and interesting results also. On the other hand, some specific improvements should be done.
Graphical abstract can be formed.
The abstract section should be rewritten according to the writing rule.
A figure can be formed for the sample preparation process.
In the 3.3.1 section, there were not any literature results.
The result of the study should be compared with the literature.
The data in tables should be aligned.
the conclusion section should be enlargement with future perspectives.
Author Response
感谢您的评论!
请参阅附件。

Reviewer 2 Report
Comments and Suggestions for Authors
Dear Authors,
I feel that the manuscript is well organized, I suggest these revisions:
Line 173-174: Has the method of extraction of volatile molecules been developed by the authors? If so, data on the reproducibility of the method should be provided
Line 175: Explain the characteristics of the column used
Indicate replicates for each chemical and sensory analysis
Line 97: What quantitative rating scale was used?
Reviewer 3 Report
Comments and Suggestions for Authors
In their manuscript, Wang et al. explored the impact of leaf grade on the taste and aroma of shaken Hunan black tea by assessing changes in biochemical composition, such as polyphenols, catechins, amino acids, and VOCs during processing. The study is well-executed and written, presenting a substantial amount of analytical data. However, several shortcomings were identified that need addressing.
1. The type of clone and harvesting season significantly influence the quality of processed tea, yet no information about these factors is included in the manuscript.
2. Some experimental details in certain sections lack clarity and understanding. For example, the process of amino acid derivatization in HPLC analysis is not clearly explained.
3. It is noted that ACCQ is unstable in water, and using a 10% ACCQ as mobile phase A may not derivatize tea amino acids effectively, especially without acid hydrolysis. The authors are urged to provide HPLC chromatograms of amino acid analysis in the manuscript or supporting information to substantiate their claims.
4. Chemical profiling of fresh tea leaves is missing from the manuscript, which is crucial for a comprehensive understanding of chemical changes during processing.
5. Inconsistency in reporting standard deviation values, with some presented and others as SD value 0, raises questions about the number of replications for each parameter. This information should be included in the experimental section.
6. The authors mention the use of sensory evaluation with human volunteers but fail to provide an ethics statement. It is essential to confirm that proper protocols were followed to protect participants' rights and privacy. A statement of consent is also necessary to ensure participants willingly participated and allowed the use of their information.
7. Furthermore, more details about the process for producing shaken Hunan black tea should be included to enhance reader comprehension.
8. Finally, clarification is sought on the term 'ABBT black tea' mentioned in line 48.
Comments on the Quality of English LanguageMinor editing of English language required
Round 2
Reviewer 3 Report
Comments and Suggestions for Authors
The authors addressed all the queries and the manuscript is substantially improved. The manuscript is recommended for publication in its present form.